# Adversarial Feature Desensitization

**Pouya Bashivan**[1,2,*]    **Reza Bayat**[2]    **Adam Ibrahim**[2]    **Kartik Ahuja**[2]

**Mojtaba Faramarzi**[2]    **Touraj Laleh**[2]    **Blake Richards**[1,2]

**Irina Rish**[2,*]

[1] McGill University, Montreal, Canada
[2] MILA, Université de Montréal, Montreal, Canada
*Correspondence to: {`bashivap,irina.rish`}`@mila.quebec`

## Abstract

Neural networks are known to be vulnerable to adversarial attacks – slight but carefully constructed perturbations of the inputs which can drastically impair the network's performance. Many defense methods have been proposed for improving robustness of deep networks by training them on adversarially perturbed inputs. However, these models often remain vulnerable to new types of attacks not seen during training, and even to slightly stronger versions of previously seen attacks. In this work, we propose a novel approach to adversarial robustness, which builds upon the insights from the domain adaptation field. Our method, called *Adversarial Feature Desensitization (AFD)*, aims at learning features that are invariant towards adversarial perturbations of the inputs. This is achieved through a game where we learn features that are both predictive and robust (insensitive to adversarial attacks), i.e. cannot be used to discriminate between natural and adversarial data. Empirical results on several benchmarks demonstrate the effectiveness of the proposed approach against a wide range of attack types and attack strengths. Our code is available at `https://github.com/BashivanLab/afd`.

## 1   Introduction

When training a classifier, it is common to assume that the training and test samples are drawn from the same underlying distribution. In adversarial machine learning, however, this assumption is intentionally violated by using the classifier itself to perturb the samples from the original (natural) data distribution towards a new distribution over which the classifier's error rate is increased [52]. As expected, when tested on such adversarially generated input distribution, the classifier severely underperforms. To date, various methods have been proposed to defend the neural networks against adversarial attacks [34, 2], additive noise patterns and corruptions [24, 25, 45], and transformations [17]. Among these methods, two of the most successful adversarial defense methods to date are adversarial training [34], which trains the neural network with examples that are perturbed to maximize the loss on the target model, and TRADES [57], which regularizes the classifier to push the decision boundary away from the data. While past adversarial defence methods have successfully improved the neural network robustness against adversarial examples, it has also been shown that these robust networks remain susceptible to even slightly larger adversarial perturbations or other forms of attacks [19, 46, 48].

In this paper, we propose to view the problem of adversarial robustness through the lens of domain adaptation, and to consider distributions of natural and adversarial images as distinct input domains

that a classifier is expected to perform well on. We then focus our attention on learning features that are invariant under such domain shifts. Building upon domain adaptation literature [4], we use the classification-based $\mathcal{H}\Delta\mathcal{H}$-divergence to quantify the distance between the natural and adversarial domains. The theory of domain adaptation allows us to formulate a bound on the adversarial classification error (i.e. the error under the distribution of adversarial examples) in terms of the classification error on natural images and the divergence between the natural and adversarial features.

We further propose an algorithm for minimizing the adversarial error using this bound. For this, we train a classifier and a domain discriminator to respectively minimize their losses on the label classification and domain discrimination tasks. The feature extractor is trained to minimize the label classifier's loss and maximise the discriminator's loss. In this way, the feature extractor network is encouraged to learn features that are both predictive for the classification task and insensitive to the adversarial attacks. The proposed setup is conceptually similar to prior work in adversarial domain adaptation [18, 53], where domain-invariant features are learned through an adversarial game between the domain discriminator and a feature extractor network.

This setup is similar to the adversarial learning paradigm widely used in image generation and transformation [20, 28, 60], unsupervised and semi-supervised learning [39], video prediction [35, 31], active learning [47], and continual learning [16]. Some prior work have also considered adversarial learning to tackle the problem of adversarial examples [54, 36, 9, 8]. These methods used generative models to learn the distribution of the adversarial images[54, 36], or to learn the distribution of input gradients[9, 8]. Unlike our method which learns a discriminator function between distributions of adversarial and natural features and updates the feature extractor to reduce the discriminability of those distributions.

The main contributions of this work are as follows:

- We apply domain-adaptation theory to the problem of adversarial robustness; this allows to bound the adversarial error in terms of the error on the natural inputs and the divergence between the feature (representation) distributions of adversarial and natural domains.

- Aiming to minimize this bound, we propose a method which learns adversarially robust features that are both predictive and insensitive to adversarial attacks, i.e. cannot be used to discriminate between natural and adversarial data.

- We empirically demonstrate the effectiveness of the proposed method in learning robust models against a wide range of attack types and attack strengths, and show that our proposed approach often significantly outperforms most previous defense methods.

## 2 Related Work

There is an extensive literature on mitigating susceptibility to adversarial perturbations [34, 57, 13, 59, 3, 22, 7]. Adversarial training [34] is one of the earliest successful attempts to improve robustness of the learned representations to potential perturbations to the input pattern by solving a min-max optimization problem. TRADES [57] adds a regularization term to the cross-entropy loss which penalizes the network for assigning different labels to natural images and their corresponding perturbed images. [41] proposed an additional regularization term (local linearity regularizer) that encourages the classification loss to behave linearly around the training examples. [55, 51] proposed to regularize the flatness of the loss to improve adversarial robustness.

Our work is closely related to the domain adaptation literature in which adversarial optimization has recently gained much attention [18, 32, 53]. From this viewpoint one could consider the clean and perturbed inputs as two distinct domains for which a network aims to learn an invariant feature set. Although in our setting, i) the perturbed domain continuously evolves while the parameters of the feature network are tuned; ii) unlike the usual setting in domain-adaptation problems, here we have access to the labels associated with some samples from the perturbed (target) domain. Recent work[49] regularized the network to have similar logit values in response to clean and perturbed inputs and showed that this additional term leads to better robust generalization to unseen perturbations. Related to this, Adversarial Logit Pairing [27] increases robustness by directly matching the logits for clean and adversarial inputs. JARN [9] Another line of work is on developing certified defenses which consist of methods with provable bounds over which the network is *certified* to operate robustly [58, 56, 10]. While these approaches provide a sense of guarantee about the proposed defenses, they

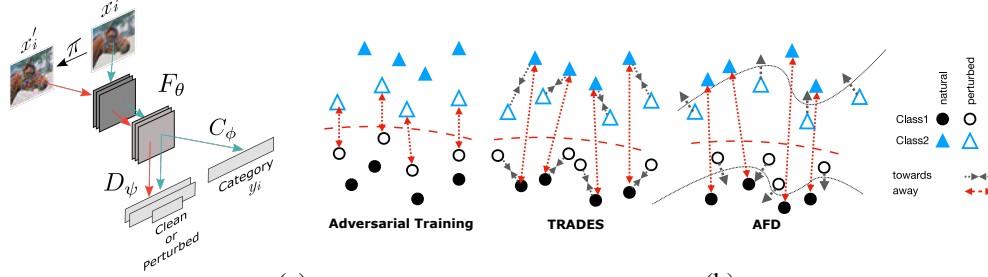

(a)                                      (b)

Figure 1:   (a) Overview of the proposed AFD approach; (b) Visual comparison of adversarial robustness methods (Adversarial training [34], TRADES [57], and AFD). The dashed red and dotted black lines correspond to the decision boundary of classes and the domain discriminator respectively.

are usually prohibitively expensive to train, drastically reduce the performance of the network on natural images, and the empirical robustness gained against standard attacks is low.

## 3   Our approach

We will now make a connection between the domain adaptation and adversarial robustness, and build upon this connection to develop an approach for improving the network's robustness against adversarial attacks.

### 3.1   Preliminaries

Let $F_\theta(x) : \mathcal{X} \to \mathcal{Z}$, where $\mathcal{X} \subseteq \mathbb{R}^n$, $\mathcal{Z} \subseteq \mathbb{R}^m$, be a *feature extractor* (e.g. a neural network with parameters $\theta$) mapping the input $x \in \mathcal{X}$ into the feature vector (representation) $z \in \mathcal{Z}$, and let $C_\phi : \mathcal{Z} \to \mathcal{Y}$, where $\mathcal{Y} = \{1, \ldots, K\}$ are the class labels, be a *classifier*, with parameters $\phi$ (e.g., the last linear layer of a neural network plus the softmax function, on top of the extracted features).

**Adversarial attack**: Let $\pi(x, \epsilon)$ denote a perturbation function (an adversarial attack) which, for a given $(x, y) \in \mathcal{X} \times \mathcal{Y}$, generates a perturbed sample $x' \in \mathcal{B}(x, \epsilon)$ within the $\epsilon$-neighborhood of $x$, $\mathcal{B}(x, \epsilon) = \{x' \in \mathcal{X} : \|x' - x\| < \epsilon\}$, by solving the following maximization problem

$$\max_{t \in \mathcal{B}(x, \epsilon)} \mathcal{L}(C_\phi(F_\theta(t)), y), \tag{1}$$

where $\mathcal{L}$ is the task classification loss function. In practice, however, the perturbed sample $x'$ found by an attacker is typically an approximate rather than the exact solution to this maximization problem.

In order to characterize the distance between the natural and adversarial data distributions, the following notion of distance between two probability distributions, defined in [4, 18], will be used later to make a connection with domain adaptation theory.

$\mathcal{H}\Delta\mathcal{H}$**-distance:** Let $\mathcal{H}$ be a set of binary classifiers (hypotheses), called a hypothesis space; then the symmetric difference hypothesis space $\mathcal{H}\Delta\mathcal{H}$ defines the set of hypotheses that capture the disagreements between two hypotheses in $\mathcal{H}$, as in [4]:

$$g \in \mathcal{H}\Delta\mathcal{H} \iff g(x) = h(x) \oplus h'(x) \quad \text{for some } h, h' \in \mathcal{H}, \tag{2}$$

where $\oplus$ denotes the XOR function. Then the $\mathcal{H}\Delta\mathcal{H}$-distance [4, 18] between two data distributions (domains) $\mathcal{S}$ and $\mathcal{T}$, with respect to the hypothesis space $\mathcal{H}$, is defined as:

$$d_{\mathcal{H}\Delta\mathcal{H}}(\mathcal{S}, \mathcal{T}) = 2 \sup_{h \in \mathcal{H}\Delta\mathcal{H}} |P_{x \sim \mathcal{S}}[h(x) = 1] - P_{x \sim \mathcal{T}}[h(x) = 1]|. \tag{3}$$

This equation turns into an inequation when the supremum is taken over the hypothesis space $\mathcal{H}$ instead of $\mathcal{H}\Delta\mathcal{H}$ [18].

### 3.2   A Domain Adaptation View of Adversarial Robustness

A *domain* is defined as a data distribution $\mathcal{D}$ on the set of inputs $\mathcal{X}$ [5]. In the adversarial robustness setting, we consider two domains – the natural and the adversarial domains, corresponding respectively

to the source and target domains in domain adaptation. We denote by $\mathcal{D}_\mathcal{X}$ and $\mathcal{D}'_\mathcal{X}$ the natural and adversarial distributions of input instances respectively and by $\mathcal{D}_\mathcal{Z}$ and $\mathcal{D}'_\mathcal{Z}$ their corresponding induced distributions over the feature space $\mathcal{Z}$. As in domain adaptation, we assume that $f : \mathcal{X} \to \mathcal{Y}$ is a labeling function common to both domains. The expected classification error $\epsilon_\mathcal{Z}$ of the classifier $C_\phi$ over $\mathcal{D}_\mathcal{Z}$ is defined as the probability that the classifier $C_\phi$ disagrees with the function $\tilde{f}$:

$$\epsilon_\mathcal{Z}(C_\phi) = E_{z \sim \mathcal{D}_\mathcal{Z}} \big[ y \neq C_\phi(z) \big], \tag{4}$$

where $\tilde{f} : \mathcal{Z} \to \mathcal{Y}$ is a mapping from the features to the class label such that $f(x) = \tilde{f}(F_\theta(x))$. We similarly define $\epsilon'_\mathcal{Z}$ as the expected error of $C_\phi$ over $\mathcal{D}_{\mathcal{Z}'}$. Using theorem 2 from [4] that relates the source and the target domain errors, we get an upper bound on the expected adversarial error $\epsilon'_\mathcal{Z}$ as:

$$\epsilon'_\mathcal{Z}(h) \leq \epsilon_\mathcal{Z}(h) + \frac{1}{2} d_{\mathcal{H}\Delta\mathcal{H}}(\mathcal{D}_\mathcal{Z}, \mathcal{D}'_\mathcal{Z}) + c, \tag{5}$$

where $c$ is a constant term w.r.t. $h$. Eq. 5 essentially gives a bound on the adversarial error $\epsilon'_\mathcal{Z}$ in terms of the natural error $\epsilon_\mathcal{Z}$ and a divergence $d_{\mathcal{H}\Delta\mathcal{H}}$ between the natural and adversarial domains with respect to their induced representation distributions $\mathcal{D}_\mathcal{Z}$ and $\mathcal{D}'_\mathcal{Z}$. In the next section, we will describe an algorithm for improving adversarial robustness of a model by iteratively estimating and minimizing these two components of the error bound.

### 3.3 Adversarial Feature Desensitization

Based on Eq. 5, the expected adversarial error could be reduced by jointly minimizing the natural error and the divergence between the distributions of natural and adversarial representations $d_{\mathcal{H}\Delta\mathcal{H}}(\mathcal{D}_\mathcal{Z}, \mathcal{D}'_\mathcal{Z})$. While minimizing the natural error $\epsilon_\mathcal{X}$ is straightforward, minimizing the cross-domain divergence requires us to estimate $d_{\mathcal{H}\Delta\mathcal{H}}(\mathcal{D}_\mathcal{Z}, \mathcal{D}'_\mathcal{Z})$. As was shown before [18], training a *domain discriminator* $D_\psi$ is closely related to estimating the $d_{\mathcal{H}\Delta\mathcal{H}}(\mathcal{D}_\mathcal{Z}, \mathcal{D}'_\mathcal{Z})$. The domain discriminator is a classifier trained to assign a label of 1 to samples from $\mathcal{D}_\mathcal{Z}$, and -1 to samples from $\mathcal{D}'_\mathcal{Z}$. Namely, it is shown [18] that

$$d_{\mathcal{H}\Delta\mathcal{H}}(\mathcal{D}_\mathcal{Z}, \mathcal{D}'_\mathcal{Z}) \leq 2 \sup_{h \in \mathcal{H}} |\alpha_{\mathcal{D}_\mathcal{Z}, \mathcal{D}'_\mathcal{Z}}(h) - 1|, \tag{6}$$

where $\alpha_{\mathcal{D}_\mathcal{Z}, \mathcal{D}'_\mathcal{Z}}(h) = P_{z \sim \mathcal{D}_\mathcal{Z}}\big[h(z) = 1\big] + P_{z \sim \mathcal{D}'_\mathcal{Z}}\big[h(z) = -1\big]$ combines the true positives and true negatives, and is thus maximized by the optimal domain discriminator $h = D_\psi$. Note that, if the domain distributions $\mathcal{D}_\mathcal{Z}$ and $\mathcal{D}'_\mathcal{Z}$ are the same, then even the best choice of domain discriminator $D_\psi$ will achieve chance-level accuracy, corresponding to $\alpha_{\mathcal{D}_\mathcal{Z}, \mathcal{D}'_\mathcal{Z}}(D_\psi) = 1$. *Our approach will aim at minimizing this estimated distance $d_{\mathcal{H}\Delta\mathcal{H}}(\mathcal{D}_\mathcal{Z}, \mathcal{D}'_\mathcal{Z})$ by tuning the feature extractor network parameters $\theta$ in the direction that pushes the distributions $\mathcal{D}_\mathcal{Z}$ and $\mathcal{D}'_\mathcal{Z}$ closer together.* In parallel, we train the domain discriminator to estimate and guide the progress of the feature extractor's tuning.

We now describe the proposed approach (see Algorithm 1) which essentially involves simultaneous training of the feature extractor $F_\theta$, the task classifier $C_\phi$ and the domain discriminator $D_\psi$ (see Figure 1a)[1]. One iteration of the training procedure consists of the following three steps.

First, parameters of the feature extractor $F_\theta$ and classifier $C_\phi$ are updated aiming to minimize the natural error $\epsilon_\mathcal{X}$ using the cross-entropy loss on natural inputs:

$$\mathcal{L}_C = -\frac{1}{m} \sum_{i=1}^{m} \tilde{y}_i \cdot \log \Big( \text{softmax}(C_\phi(F_\theta(x_i))) \Big), \tag{7}$$

where $\tilde{y}_i$ is a one-hot encoding of the true label of the $i$-th sample $x_i$.

Next, steps two and three essentially implement a two-player minimax game similar to that in Generative Adversarial Networks (GAN) [20], carried out between the feature extractor network $F_\theta$ and the domain discriminator $D_\psi$, with a value function

$$V(F_\theta, D_\psi) = \mathbb{E}_{p(y)} \big[ \mathbb{E}_{p(x|y)} [\mathcal{S}(-D_\psi(F_\theta(x), y))] \big] + \mathbb{E}_{q(y)} \big[ \mathbb{E}_{q(x|y)} [\mathcal{S}(D_\psi(F_\theta(x), y))] \big], \tag{8}$$

---

[1]Note that we will somewhat abuse the notation, assuming that $C_\phi$ and $D_\psi$ below correspond to the logits (last-layer output) of the corresponding networks. Also, we will use class-conditional discriminators, $D_\psi(F_\theta(x, y))$, i.e. train different domain discriminator for different label values $y$.

**Algorithm 1:** AFD training procedure

---

**Input:** Adversarial perturbation function (attack) $\pi$, feature extractor $F_\theta$, task classifier $C_\phi$, domain discriminator $D_\psi$, learning rates $\alpha$, $\beta$, and $\gamma$.

**repeat**

    input next mini-batch $\{(x_i, y_i), ..., (x_m, y_m)\}$

    for i=1 to m: $x'_i \leftarrow \pi(x_i, \epsilon)$

    Compute $\mathcal{L}_C$ according to Eq. 7

    Compute $\mathcal{L}_D$ according to Eq. 9

    Compute $\mathcal{L}_F$ according to Eq. 10

    $(\theta, \phi) \leftarrow (\theta, \phi) - \alpha\nabla_{\theta, \phi}\mathcal{L}_C$     % update feature extractor and task classifier

    $\psi \leftarrow \psi - \beta\nabla_\psi\mathcal{L}_D$     % update domain discriminator

    $\theta \leftarrow \theta - \gamma\nabla_\theta\mathcal{L}_F$     % update feature extractor

**until** *convergence*;

---

where $\mathcal{S}$ is the softplus function. In particular, parameters of the domain discriminator $D_\psi$ are updated to minimize the cross-entropy loss associated with discriminating natural and adversarial inputs, maximizing $\alpha(h)$ in Eq. 6.

$$\mathcal{L}_D = \frac{1}{m}\sum_{i=1}^{m}\Big[\mathcal{S}(-D_\psi(F_\theta(x_i), y_i)) + \mathcal{S}(D_\psi(F_\theta(x'_i), y_i))\Big], \tag{9}$$

while the parameters of the feature extractor function $F_\theta$ are adversarially updated to maximize the domain discriminator's loss from Eq. 9

$$\mathcal{L}_F = \frac{1}{m}\sum_{i=1}^{m}\mathcal{S}(-D_\psi(F_\theta(x'_i), y_i)). \tag{10}$$

In Figure 1b, we visually compare the learning dynamics in adversarial training, TRADES and AFD. Essentially, the adversarial training solves the classification problem by pushing the representation of adversarial examples from different classes away. TRADES regularizes the normal classification loss on the natural inputs with an additional term that encourages the representation of adversarial and natural images to match. Similar to TRADES, in AFD, the regular classification loss on natural inputs is augmented but with an adversarial game which consists of training the domain discriminator that distinguishes between the adversarial and natural inputs for each class followed by updates to the feature extractor to make the representations for natural and adversarial examples to become indistinguishable from each other. Notably, because the parameter update for the feature extractor network is done to maximize the domain discriminator loss and not to decrease the loss for particular adversarial examples (as is done in adversarial training or TRADES), it potentially increases the network robustness against any perturbation that could be correctly classified using the same domain discriminator. This could potentially lead to a broader form of generalization learned by the network.

**Discussion: Relation to Adversarial Training.** Adversarial training minimizes the expected error on adversarial examples (the perturbed versions of the natural samples), generated by an attacker in order to maximize the classification loss. The adversarial training procedure involves a minimax optimization problem consisting of an inner maximization to find adversarial examples that maximize the classification loss and an outer minimization to find model parameters that minimize the adversarial loss. From the domain adaptation point of view, the inner optimization of adversarial training is equal to a sampling procedure that generates samples from the target domain. Intuitively, direct training of the classifier on samples from the target domain would be the best way to improve the accuracy in that domain (i.e. adversarial classification accuracy). However, it's important to note that the adversarial examples found through the inner optimization only approximately maximize the classification loss, and therefore the adversarial error associated with these samples only act as a lower bound on the true adversarial error and therefore the outer loop of the adversarial training method essentially minimizes a lower bound on the adversarial classification error. In contrast to this setup, our proposed method minimizes a conservative upper bound on the adversarial error and therefore is more likely to generalize to a larger set of unseen attacks, and to stronger versions of previously seen attacks (i.e. ones that generate higher-loss samples in the inner optimization loop).

## 4 Experiments

### 4.1 Experimental setup

**Datasets.** We validated our proposed method on several common datasets including MNIST [30], CIFAR10, CIFAR100 [29], and tiny-Imagenet [26]. The inputs for all datasets were used in their original resolution except for tiny-Imagenet where the inputs were resized to $32 \times 32$ to allow the experiments to finish within reasonable time on two GPUs.

**Adversarial attacks.** To fairly assess the generalization ability of each defense method across attack types, we tested each network on 9 well-known adversarial attacks from the literature, using existing implementations from the Foolbox [42] and Advertorch [12] Python packages. Namely, we tested the models against different variations of the Projected Gradient Descent (PGD) [34] ($L_\infty$, $L_2$, $L_1$), Fast Gradient Sign Method (FGSM) [21], Momentum Iterative Method (MIM) [14], Decoupled Direction and Norm (DDN) [43], Deepfool [40], C&W [6], and AutoAttack [11] attacks. Also to assess the generalization in robustness across stronger adversarial attacks, for each attack we also varied the $\epsilon$ value across a wide range and validated different models on each. Specific hyperparameters used for each attack are listed in Table-A2.

**Feature extractor network $F_\theta$ and classifier $C_\phi$.** We used the same network architecture, ResNet18 [23] for the feature extractor and classifier networks in experiments on all datasets and only increased the number of features for more challenging datasets. The number of base filters in the ResNet architecture was set to 16 for MNIST and 64 for other datasets. We used the activations before the last linear layer as the the output of the feature extractor network ($\mathcal{Z}$) and the last linear layer as the classifier network $C_\phi$. We added an activation normalization layer to the output of feature extractor network to provide normalized features to both $C_\theta$ and $D_\psi$ networks.

**Domain discriminator network $D_\psi$.** We compared several variations of the domain discriminator architecture and evaluated its effect on robust classification on MNIST dataset (Table A5). Overall, we found that using deeper networks for domain discriminator and adding projection discriminator layer improves the robust classification accuracy. The number of hidden units in all layers of $D_\psi$ were equal (64 for MNIST and 512 for other datasets). Following the common design principles in Generative Adversarial Networks literature, we used the spectral normalization [37] on all layers of $D_\psi$. In all experiments, the domain discriminator ($D_\psi$) consisted of three fully connected layers with Leaky ReLU nonlinearity followed by a projection discriminator layer that incorporated the labels into the adversarial discriminator through a dot product operation [38]. Further details of training for each experiment are listed in Table-A1.

**Training parameters and baselines.** All networks including baselines were trained on an adaptive version of PGD attack [11] that adaptively tunes the step size during the attack with virtually no computational overhead compared to standard PGD attack. We used $\epsilon = 0.3$, 0.031, and 0.016 for MNIST, CIFAR, and Tiny-Imagenet datasets respectively. To find the best learning rates, we randomly split the CIFAR10 train set into a train and validation sets (45000 and 5000 images in train and validation sets respectively). We then carried out a grid-search using the train-validation sets and picked the learning rates with highest validation performance. Based on this analysis, we selected the learning rate $\gamma = 0.5$ for tuning the feature extractor $F_\theta$, and $\alpha = \beta = 0.1$ for tuning the parameters in domain discriminator $D_\psi$, and the task classifier $C_\phi$.

In all experiments we trained two versions of the AFD model, one with losses $L_D$ and $L_F$ according to Eq. 9 and 10 which we call AFD-DCGAN and another version where we substitute the losses with those from the Wasserstein GAN [1] dubbed AFD-WGAN (see Eq. 11 and 12 in the Appendix). We mainly compared the performance of our proposed method with two prominent defense methods, adversarial training and TRADES. We used a re-implementation of adversarial training (AT) method [34] and the official code for TRADES[2] [57] and denoted these results with † in the tables. All experiments were run on NVIDIA V100 GPUs. We used one GPU for experiments on MNIST and 2 GPUs for other datasets.

### 4.2 Robust classification against nominal attacks

We first evaluated our method against adversarial attacks under similar settings to those used during training ($\epsilon = 0.3$, 0.031, and 0.015 for MNIST, CIFAR, and Tiny-Imagenet datasets respectively).

---

[2]`https://github.com/yaodongyu/TRADES.git`

Table 1: Comparison of adversarial accuracy against various attacks on different datasets. For $PGD_\infty$ attack we used $\epsilon = 0.3$, $0.031$, and $0.015$ for MNIST, CIFAR10/CIFAR100, and Tiny-Imagenet datasets respectively and for C&W attack we used $\epsilon = 1$ for all datasets. † indicates replicated results. NT: natural training; AT: adversarial training; AFD: adversarial feature desensitization; WB: white-box attack; BB: black-box attack where the adversarial examples were produced by running the attack on the NT ResNet18 model. Numbers reported with $\mu \pm \sigma$ denote mean and std values over three independent runs with different random initialization. * RST[7] additionally uses 500K unlabeled images during training.

| Method | Dataset | Network | Clean | $\textbf{PGD}_\infty$ (WB) | $\textbf{C\&W}_2$ (WB) | $\textbf{AA}_\infty$ (WB) | $\textbf{PGD}_\infty$ (BB) | $\textbf{C\&W}_2$ (BB) |
|---|---|---|---|---|---|---|---|---|
| NT† | | RN18 | 98.84 | 0. | 62.43 | 0.0 | 50.82 | 96.48 |
| AT[34]† | | RN18 | **99.35** | 95.66 | 92.78 | 89.99 | **98.92** | **98.95** |
| TRADES[57]† | MNIST | RN18 | 99.14 | 94.81 | 90.08 | 88.66 | 98.5 | 98.57 |
| AFD-DCGAN | | RN18 | 99.24 | 95.72 | 93.78 | 88.79 | 98.65 | 98.49 |
| AFD-WGAN | | RN18 | 99.14 | **97.68** | **97.68** | **90.12** | 98.59 | 98.71 |
| AT[34] | | RN18 | 87.3 | 45.8 | - | - | 86.0 | - |
| TRADES[57] | | RN18 | 84.92 | 56.61 | - | - | 87.60 | - |
| RLFAT[50] | | WRN-32-10 | 82.72 | 58.75 | - | - | - | - |
| RST+[55, 7]* | | WRN-34-10 | 89.82 | 64.86 | - | - | - | - |
| LLR[41] | | WRN-28-8 | **86.83** | 52.99 | - | - | - | - |
| JARN[9] | CIFAR10 | WRN-34-10 | 84.8 | 46.7 | - | - | - | - |
| NT† | | RN18 | 94.89 | 0.55 | 0.31 | 0.0 | 17.93 | - |
| AT[34]† | | RN18 | 85.92 | 40.07 | 40.27 | 36.14 | 85.14 | 85.84 |
| TRADES[57]† | | RN18 | 81.94 | 53.3 | 40.24 | **43.48** | 80.82 | 81.74 |
| AFD-DCGAN | | RN18 | 86.82 | 44.35 | 50.93 | 34.46 | **85.73** | **86.68** |
| AFD-WGAN | | RN18 | 85.95 | **59.38** | **62.43** | 37.33 | 84.74 | 85.79 |
| NT† | | RN18 | 76.76 | 0.01 | 0.52 | 0.02 | - | - |
| AT[34]† | | RN18 | 56.49 | 18.54 | 17.71 | 18.30 | 56.07 | 56.42 |
| TRADES[57]† | CIFAR100 | RN18 | 60.32 | **25.11** | 20.52 | **21.10** | 59.62 | 60.29 |
| AFD-DCGAN | | RN18 | **60.95** | 18.06 | 21.47 | 16.31 | **60.31** | **60.86** |
| AFD-WGAN | | RN18 | 58.87 | 22.35 | **25.33** | 18.00 | 58.15 | 58.75 |
| NT† | | RN18 | 58.30 | 0.3 | 0.0 | 0.0 | - | - |
| AT[34]† | Tiny-IN | RN18 | 43.80 | 12.62 | 4.90 | 9.48 | 41.87 | 42.86 |
| TRADES[57]† | | RN18 | 37.70 | **13.26** | 4.11 | **12.57** | 36.26 | 36.72 |
| AFD-WGAN | | RN18 | **47.70** | 11.49 | **5.90** | 9.45 | **43.5** | **44.69** |

Table 1 compares the robust classification performance of AFD and several other defense methods against PGD-$L_\infty$, C&W-$L_2$ and AutoAttack white-box and black-box attacks. The black-box attacks were carried out by constructing the adversarial examples using a ResNet18 architecture trained on the natural inputs $x \sim D_\mathcal{X}$. Overall both versions of AFD (AFD-DCGAN and AFD-WGAN) were highly robust against all five tested attacks while maintaining a higher "Clean" accuracy (on natural data) compared to strong baseline models like TRADES and Adversarial Training. AFD-WGAN was consistently at the top on MNIST and CIFAR10 datasets. On CIFAR100 and Tiny-Imagenet, AFD performed better than or similar to Adversarial Training on all the attacks and performed better than TRADES on most of the attacks, although it was occasionally behind TRADES (on PGD-$L_\infty$ and AA white-box attacks). Analysis of feature sensitivity showed that on MNIST and CIFAR10 datasets on which AFD outperformed the other baselines by a larger margin, the features were significantly more insensitive to adversarial perturbations and over a larger range of attack strengths (Figure-A4). In addition to these tests, we also evaluated the AFD model against transfer black-box attacks from Adversarial Training and TRADES models which further demonstrated AFD's higher robustness to those attacks too (Table-A3).

### 4.3 Robust classification against stronger and unseen attacks

To evaluate how each network generalizes to unseen domains of adversarial inputs (i.e. adversarial attacks generated with unseen forms of adversarial attacks), we additionally validated the classification robustness against a range of possible $\epsilon$ values for several widely used attacks that were not used during training. To fairly compare different models while considering both attack types and $\epsilon$ values, we computed the area-under-the-curve (AUC) for accuracy vs. epsilon for each attack (similar to Figure-2). Table-2 summarizes the AUC values for all 9 attack methods on four tested datasets. Compared with the baselines, we found that, AFD-trained networks consistently performed better on various datasets and on almost all the tested attacks even for substantially larger $\epsilon$ values (Figure 2, also see Figures A1,A3 in the appendix). These results show that compared to other baselines, AFD-trained networks are robust against a wider range of attacks and attack strengths ($\epsilon$). This further

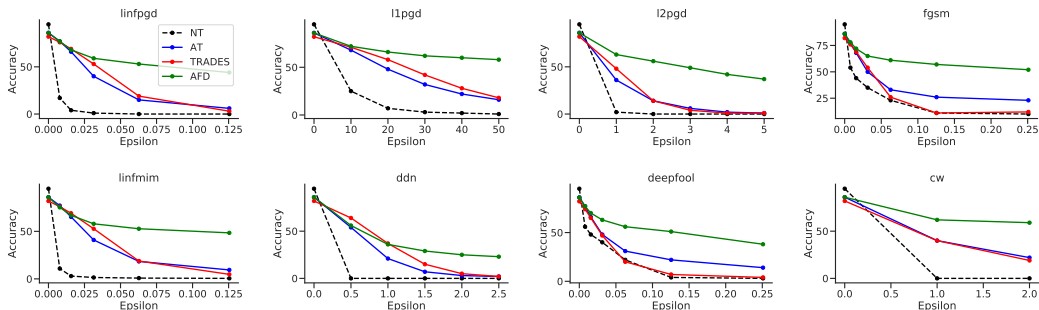

Figure 2: Adversarial accuracy for different methods against white-box attacks on CIFAR10 dataset with ResNet18 architecture.

Table 2: AUC measures for different perturbations and methods on MNIST, CIFAR10, CIFAR100, and tiny-imagenet datasets. AUC values are normalized to have a maximum allowable value of 1. Evaluations on AT and TRADES were made on networks trained using reimplemented or official code.

| Dataset | Model | $PGD_{L_\infty}$ | $PGD_{L2}$ | $PGD_{L1}$ | FGSM | MIM | DDN | DeepFool | C&W | AA |
|---------|-------|------|------|------|------|------|------|------|------|------|
| MNIST | NT | 0.16 | 0.06 | 0.07 | 0.3 | 0.19 | 0.09 | 0.21 | 0.57 | 0.28 |
| | AT | 0.74 | 0.29 | 0.19 | 0.83 | 0.95 | 0.49 | 0.55 | 0.87 | 0.89 |
| | TRADES | 0.71 | 0.26 | 0.15 | 0.79 | 0.88 | 0.42 | 0.47 | 0.86 | 0.88 |
| | AFD-DCGAN | 0.77 | 0.33 | 0.3 | 0.78 | 0.91 | 0.51 | 0.49 | 0.9 | 0.88 |
| | AFD-WGAN | **0.92** | **0.54** | **0.55** | **0.9** | **0.98** | **0.68** | **0.63** | **0.94** | **0.90** |
| CIFAR10 | NT | 0.05 | 0.1 | 0.17 | 0.19 | 0.05 | 0.1 | 0.16 | 0.1 | 0.12 |
| | AT | 0.28 | 0.2 | 0.44 | 0.33 | 0.31 | 0.26 | 0.29 | 0.31 | 0.22 |
| | TRADES | 0.32 | 0.22 | 0.5 | 0.24 | 0.32 | 0.33 | 0.18 | 0.28 | **0.25** |
| | AFD-DCGAN | 0.34 | **0.54** | 0.43 | 0.4 | 0.31 | **0.4** | 0.43 | 0.47 | 0.22 |
| | AFD-WGAN | **0.56** | **0.54** | **0.66** | **0.59** | **0.56** | **0.4** | **0.52** | **0.62** | 0.24 |
| CIFAR100 | NT | 0.03 | 0.08 | 0.1 | 0.07 | 0.03 | 0.08 | 0.06 | 0.08 | 0.09 |
| | AT | 0.13 | 0.1 | 0.24 | 0.13 | 0.14 | 0.14 | 0.12 | 0.15 | 0.13 |
| | TRADES | 0.16 | 0.13 | 0.31 | 0.12 | 0.17 | **0.18** | 0.1 | 0.16 | **0.15** |
| | AFD-DCGAN | 0.14 | 0.12 | 0.27 | 0.17 | 0.16 | 0.15 | 0.16 | 0.18 | 0.13 |
| | AFD-WGAN | **0.18** | **0.16** | **0.31** | **0.22** | **0.19** | 0.16 | **0.19** | **0.23** | 0.13 |
| Tiny-IN | NT | 0.04 | 0.03 | 0.08 | 0.05 | 0.04 | 0.06 | 0.07 | 0.07 | 0.07 |
| | AT | **0.10** | 0.03 | 0.16 | **0.15** | **0.09** | 0.14 | 0.13 | 0.11 | 0.14 |
| | TRADES | **0.10** | 0.03 | 0.16 | 0.07 | **0.09** | **0.15** | 0.11 | 0.09 | **0.16** |
| | AFD-WGAN | **0.10** | **0.04** | **0.19** | 0.12 | **0.09** | **0.15** | **0.16** | **0.12** | 0.15 |

suggests that the features learned through AFD generalize better across various forms of attacks and can sustain larger perturbations.

We also observed that the AFD-WGAN performs better than AFD-DCGAN under most tested conditions. This is potentially due to: 1) WGAN's ability to avoid vanishing gradients when the discriminator becomes too good compared to the generator (the feature extractor function in our case) [5]; 2) WGAN's ability to avoid mode-collapses during training. In training GANs, mode collapses lead to the generator network to only output a limited set of patterns instead of learning to produce a diverse set of natural-looking images that fool the discriminator. Under our setting, WGAN potentially leads to learning a feature extractor that can produce a more diverse set of features for perturbed inputs, instead of focusing on a subset of latent dimensions. This suggests that applying more advanced GAN training algorithms could potentially further improve the robust performance in AFD-type models.

### 4.4 Estimated $\mathcal{H}\Delta\mathcal{H}$-distance and adversarial-vs-natural generalization gap

As stated in Eq. 5, the upper bound on the adversarial error can be stated in terms of the natural error, the divergence between the two domains, and a constant term. In practice, this means that the smaller the divergence term $d_{\mathcal{H}\Delta\mathcal{H}}$ is, the smaller the gap between the adversarial and natural errors ($\epsilon'_Z - \epsilon_Z$) can be. We empirically tested this prediction using the domain discriminator trained on CIFAR10 dataset using the PGD-$L_\infty$ attack. Figure-3a shows that the estimated $d_{\mathcal{H}\Delta\mathcal{H}}$ using the domain discriminator (i.e., using the corresponding empirical value of $\alpha$ in Eq. 6) trained on $PGD - L_\infty$ with $\epsilon = 0.031$ is closely related to the adversarial-vs-natural generalization gap over

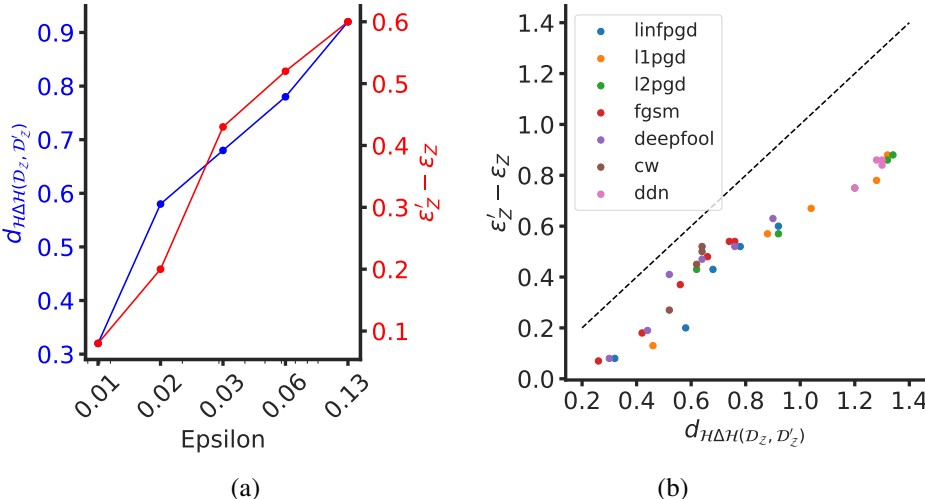

(a)             (b)

Figure 3: (a) Estimated $d_{\mathcal{H}\Delta\mathcal{H}}$ distance (based on empirical value of $\alpha$ in Eq. 6) and generalization gap in adversarial and natural error $\epsilon'_Z - \epsilon_Z$ as a function of epsilon for $PGD - L_\infty$ attack; (b) scatter plot of the estimated $d_{\mathcal{H}\Delta\mathcal{H}}$ distance using the domain discriminator and the gap in adversarial and natural error across different attack types and magnitudes (i.e. $\epsilon$). Colors correspond to different attack types. Each dot corresponds to one attack evaluated at a particular epsilon value. Estimations of the $d_{\mathcal{H}\Delta\mathcal{H}}$ distance for all attacks and epsilons are made with the domain discriminator trained on PGD-$L_\infty$ with $\epsilon = 0.031$.

different $\epsilon$ values as predicted by Eq. 5. Moreover, estimations from the same domain discriminator also predicts the gap in generalization error attained for other forms of attacks (even ones not seen during AFD training) and $\epsilon$ values with high accuracy (Figure-3b). This further supports the proposal that minimizing the estimated distance between the natural and adversarial representations can be an efficient way to improve the model robustness against various adversarial attacks.

### 4.5    Learning a sparse representation

Because the AFD method aims to learn a representation that is insensitive to adversarial attacks, we expected the learned representational space to potentially be of lower dimensionality (i.e. less number of orthogonal features). To test this, we compared the dimensionality of the learned representation using two measures. i) number of non-zero features over the test set within each dataset and ii) number of Principal Component Analysis (PCA) dimensions that explains more than 99% of the variance in the representation computed over the test-set of each dataset. We found that the same network architecture (i.e. ResNet18), when trained with AFD method learns a much sparser and lower dimensional representational space (Table A4) compared to the naturally trained, adversarial training and TRADES models. The representational spaces learned with AFD on MNIST, CIFAR10, and CIFAR100 datasets had only 6, 9, and 76 principal components respectively.

### 4.6    Adversarial and norm-based desensitization

To investigate whether the same level of robustness could be achieved by encouraging the network to produce similar representations in response to natural and adversarial inputs, we ran an additional experiment on the MNIST dataset in which we added a regularization term to the classification loss to directly minimize the representation sensitivity $S_e = \frac{1}{n}\sum_x \|F(x) - F(x')\|$, during training. We observed that although this augmented loss led to learning robustness representations, it achieved modest levels of robustness ($\sim 80\%$) and showed only weak generalization to stronger and other unseen attacks (Figure-A5). This result suggests that more direct forms of enforcing representational similarity may not lead to the same form of robustness with generalization properties similar to that achieved using an adversarial training with domain discriminator (e.g. as in AFD).

## 5    Conclusion and limitations

Decreasing the input-sensitivity of features has long been desired in training neural networks [15] and has been suggested as a way to improve adversarial robustness [44, 61]. In this work we proposed an

algorithm to decrease the sensitivity of neural network representations using an adversarial learning paradigm that involves joint training of a domain discriminator, a feature extractor, and a task classifier. Essentially, our proposed algorithm iteratively estimates a bound on the adversarial error in terms of the natural error and a classification-based measure of distance between the distributions of natural and adversarial features and then minimizes the adversarial error by concurrently reducing the natural error as well as the distance between the two feature distributions.

**Limitations**. The empirical results presented here suggest that AFD-trained models are robust against a wide range of adversarial attacks (distributions) even compared to strong baselines like Adversarial Training and TRADES. However, it is not guaranteed that the model would remain robust against any unseen attacks that we have not tested or may be invented in the future - as is the case in domain adaptation literature and the lack of theoretical guarantees for cross-domain generalization. With regards to the computational cost, when measuring the average per-epoch training time on the CIFAR10 dataset (using 2 NVIDIA V100 GPUs), we found that the AFD training time is 31% longer than adversarial training and only 4% longer than TRADES. This shows that while AFD requires three SGD updates per batch, the additional computational cost is not significantly higher than many prior methods when considering that most of the computational cost is associated with generating the adversarial examples during training.

# 6 Acknowledgements

We would like to thank Isabela Albuquerque, Joao Monteiro, and Alexia Jolicoeur-Martineau for their valuable comments on the manuscript. Pouya Bashivan was partially supported by the Unifying AI and Neuroscience – Québec (UNIQUE) Postdoctoral fellowship and NSERC Discovery grant RGPIN-2021-03035. Irina Rish acknowledges the support from Canada CIFAR AI Chair Program and from the Canada Excellence Research Chairs Program.

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
