# A Appendix

## A.1 Network architectures

For all experiments, we trained the ResNet18 architecture [23] using SGD optimizer with 0.9 momentum and learning rates as indicated in Table-A1, weight decay of $10^{-4}$, batch size of 128. All learning rates were reduced by a factor of 10 after scheduled epochs.

Table A1: Training hyperparameters for each dataset and network.

| Dataset | Model | $\mathbf{LR}_E$ | $\mathbf{LR}_{Da}$ | $\mathbf{LR}_{EDc}$ | weight decay | batch size | Num. Epochs | Scheduled Epochs |
|---|---|---|---|---|---|---|---|---|
| MNIST | | | | | | | 100 | [50, 80] |
| CIFAR-10 | ResNet18 | 0.5 | 0.1 | 0.1 | $10^{-4}$ | 128 | 300 | [150, 250] |
| CIFAR-100 | | | | | | | 300 | [150, 250] |
| Tiny-Imagenet | | | | | | | 500 | [300, 450] |

## A.2 Adversarial attacks

We used a range of adversarial attacks in our experiments. Hyperparameters associated with each attack are listed in the table below. Implementation of these attacks were adopted from Foolbox [42], AdverTorch [12] packages.

## A.3 Wasserstein GAN Loss

For AFD-WGAN model, we used the generator and discriminator losses from [1] to adversarially train the feature extractor $F_\theta$ and domain discriminator $D_\psi$ respectively.

$$\mathcal{L}_D = \frac{1}{m} \sum_{i=1}^{m} \Big[ - D_\psi(F_\theta(x_i), y_i) + D_\psi(F_\theta(x_i'), y_i) \Big] \tag{11}$$

$$\mathcal{L}_F = \frac{1}{m} \sum_{i=1}^{m} -D_\psi(F_\theta(x_i), y_i) \tag{12}$$

# B Broader Impact

As the application of deep neural networks becomes more common in everyday life, security and dependability of these networks becomes more crucial. While these networks excel at performing many complicated tasks under standard settings, they often are criticized for their lack of reliability under broader settings. One of the main points of criticism of today's artificial neural networks is on their vulnerability to adversarial patterns – slight but carefully constructed perturbations of the inputs which drastically decrease the network performance.

Our work presented here proposes a new way of addressing this important issue. Our approach could be used to improve the robustness of learned representation in an artificial neural network and as shown lead to a recognition behavior that is more aligned with the human judgement. More broadly, the ability to learn robust representations and behaviors is highly desired in a wide range of applications and disciplines including perception, control, and reasoning and we expect the presented work to influence the future studies in these areas.

Table A2: Attack hyperparameters for each dataset and attack.

| Attack | Dataset | Steps | $\epsilon$ | More | Toolbox |
|---|---|---|---|---|---|
| FGSM | MNIST
CIFAR
Tiny-IN | 1 | $[0, 0.1, 0.3, 0.35, 0.4, 0.45, 0.5]$
$[0, \frac{2}{255}, \frac{4}{255}, \frac{8}{255}, \frac{16}{255}, \frac{32}{255}, \frac{64}{255}]$
$[0, \frac{2}{255}, \frac{4}{255}, \frac{8}{255}, \frac{16}{255}, \frac{32}{255}]$ | -
-
- | Foolbox |
| PGD-$L_1$ | MNIST
CIFAR
Tiny-IN | 50 | $[[0, 10, 50, 100, 200]]$

$[0, 10, 50]$ | step=0.025 | Foolbox |
| PGD-$L_2$ | MNIST
CIFAR
Tiny-IN | 50 | $[0, 2, 5, 10]$

$[0, 2, 5]$ | step=0.025 | Foolbox |
| PGD-$L_\infty$ | MNIST
CIFAR
Tiny-IN | 40
20
20 | $[0, 0.1, 0.3, 0.35, 0.4, 0.45, 0.5]$
$[0, \frac{2}{255}, \frac{4}{255}, \frac{8}{255}, \frac{16}{255}, \frac{32}{255}]$
$[0, \frac{2}{255}, \frac{4}{255}, \frac{8}{255}, \frac{16}{255}]$ | step=0.033

step=$\frac{2}{255}$ | Foolbox |
| MIM | MNIST
CIFAR
Tiny-IN | 40 | $[0, 0.1, 0.3, 0.5, 0.8, 1]$
$[0, \frac{2}{255}, \frac{4}{255}, \frac{8}{255}, \frac{16}{255}, \frac{32}{255}]$
$[0, \frac{2}{255}, \frac{4}{255}, \frac{8}{255}, \frac{16}{255}]$ | -
-
- | AdverTorch |
| DDN | MNIST
CIFAR
Tiny-IN | 100 | $[0, 1, 2, 5]$
$[0, 2, 5, 10, 15]$
$[0, 0.2, 0.5, 1]$ | -
-
- | Foolbox |
| Deepfool | MNIST
CIFAR
Tiny-IN | 50 | $[0, 0.1, 0.3, 0.35, 0.4, 0.45, 0.5]$
$[0, \frac{2}{255}, \frac{4}{255}, \frac{8}{255}, \frac{16}{255}, \frac{32}{255}, \frac{64}{255}]$
$[0, \frac{2}{255}, \frac{4}{255}, \frac{8}{255}, \frac{16}{255}]$ | -
-
- | Foolbox |
| C&W | MNIST
CIFAR
Tiny-IN | 100 | $[0, 0.5, 1, 1.5, 2]$ | stepsize=0.05 | Foolbox |
| AA | MNIST
CIFAR
Tiny-IN | 100 | $[0, 0.2, 0.3, 0.35]$
$[0, 8/255., 16/255., 32/255.]$
$[0, 2/255., 4/255., 8/255.]$ | - | AutoAttack |

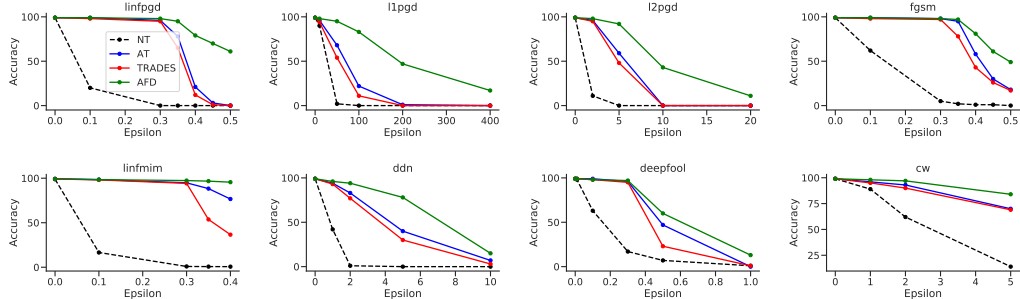

Figure A1: Comparison of adversarial accuracy of different methods against white-box attacks on MNIST dataset with ResNet18 architecture.

Table A3: Transfer black-box attack from ResNet18 network trained with adversarially training (AT) and TRADES on different datasets.

| Dataset | Method | AT Transfer | TRADES Transfer |
|---------|--------|-------------|-----------------|
| MNIST | AT | - | 97.32 |
| | TRADES | 96.64 | - |
| | AFD | **97.41** | **97.58** |
| CIFAR10 | AT | - | 64.34 |
| | TRADES | 78.43 | - |
| | AFD | **86.36** | **66.49** |
| CIFAR100 | AT | - | **42.54** |
| | TRADES | 39.22 | - |
| | AFD | **43.59** | 42.26 |

Table A4: Dimensionality of the learned representation space on various datasets using different methods and measures. Units: number of non-zero feature dimensions over the test-set within each dataset. Dims: number of PCA dimensions that account for 99% of the variance across all images within the test-set of each dataset.

| Dataset | MNIST | | CIFAR10 | | CIFAR100 | |
|---------|-------|------|---------|------|----------|------|
| Network | ResNet18 | | ResNet18 | | ResNet18 | |
| | Units | Dims | Units | Dims | Units | Dims |
| NT | 64 | **9** | 512 | 70 | 512 | 376 |
| AT | 64 | **9** | 512 | 75 | 512 | 440 |
| TRADES | 64 | 14 | 512 | 70 | 512 | 339 |
| AFD | **28** | **9** | **389** | **12** | **511** | **304** |

Table A5: Comparison of adversarial accuracy on MNIST dataset against PGD-$L_\infty$ with $\epsilon = 0.3$ for different domain discriminator architectures. FC1 and FC3 architectures refer to 1-layer and 3-layer fully connected networks respectively. PD refers to projection discriminator.

| Dataset | Model | $Da$ Architecture | Adversarial Acc. |
|---------|-------|-------------------|------------------|
| MNIST | RN18 | FC1-PD | 85.96 |
| | | FC3 | 90.73 |
| | | FC3-PD | **97.03** |

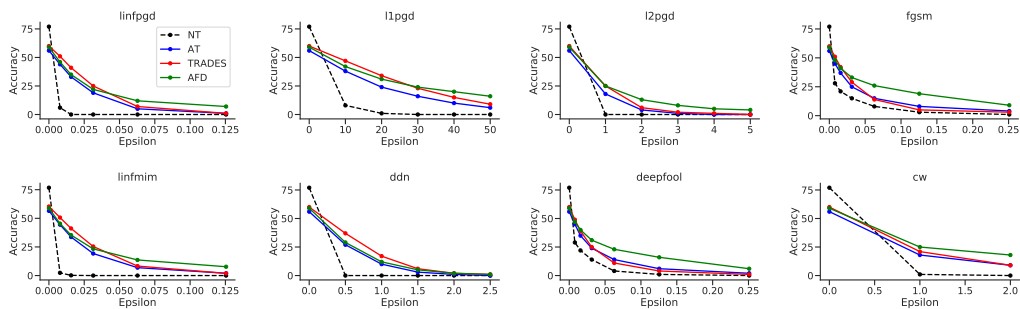

Figure A2: Comparison of adversarial accuracy of different methods against white-box attacks on CIFAR100 dataset with ResNet18 architecture.

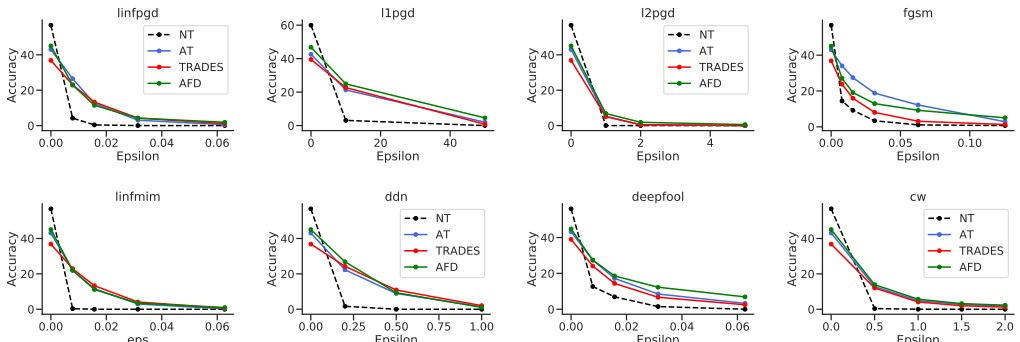

Figure A3: Comparison of adversarial accuracy of different methods against white-box attacks on Tiny-Imagenet dataset with ResNet18 architecture.

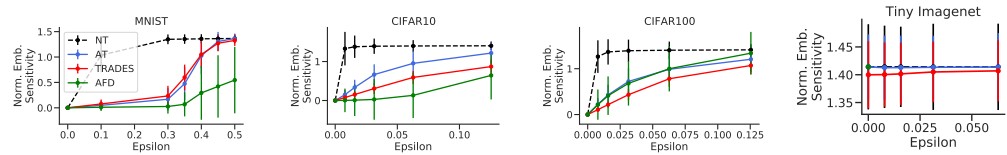

Figure A4: Comparison of normalized feature sensitivity on test set of MNIST, CIFAR10, CIFAR100, and Tiny-Imagenet datasets under PGD-$L_\infty$ attack. For each image, we computed the normalized feature sensitivity as $\frac{\|F(x)-F(x')\|_2}{\|F(x)\|_2}$. Plots show the median sensitivity over test-set of each dataset. Error bars correspond to standard deviation. (dashed-black) naturally trained; (blue) adversarially trained; (red) TRADES; (green) AFD.

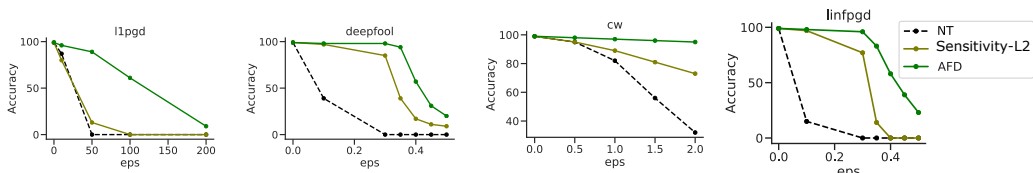

Figure A5: Comparison of adversarial accuracy of AFD and representation matching against white-box attacks on MNIST dataset with ResNet18 architecture.

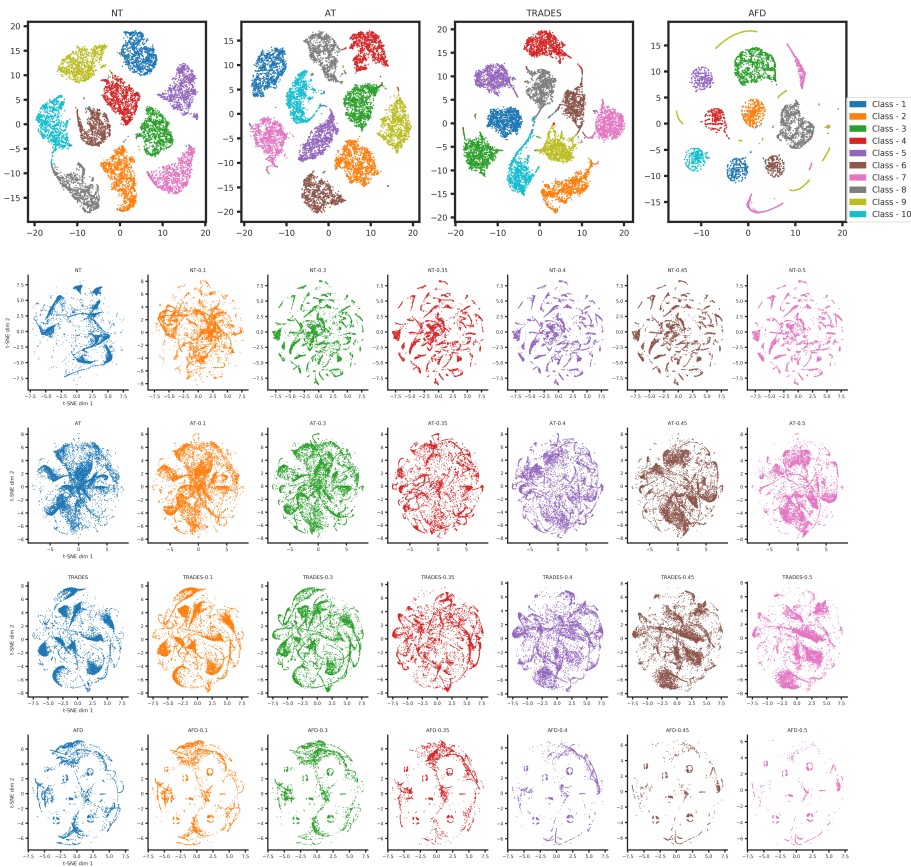

Figure A6: Scatter plot of 2-dimensional t-SNE projection [33] of the representations derived from training the ResNet18 architecture on MNIST dataset. (top row) t-SNE projection of representations of natural images for networks trained with different methods. Each point corresponds to the representation of one of the images from the MNIST test-set. (rows 2 to 5) t-SNE projection of the representation of the natural and adversarial MNIST test-set images. Columns are sorted from left to right with the strength of the perturbation (left-most column corresponds to natural images and right-most column with highest tested perturbation). Perturbations are generated using PGD-$L_\infty$ attack. NT: naturally trained; AT: adversarially trained[34]; TRADES: [57]; AFD: adversarial feature desensitization.

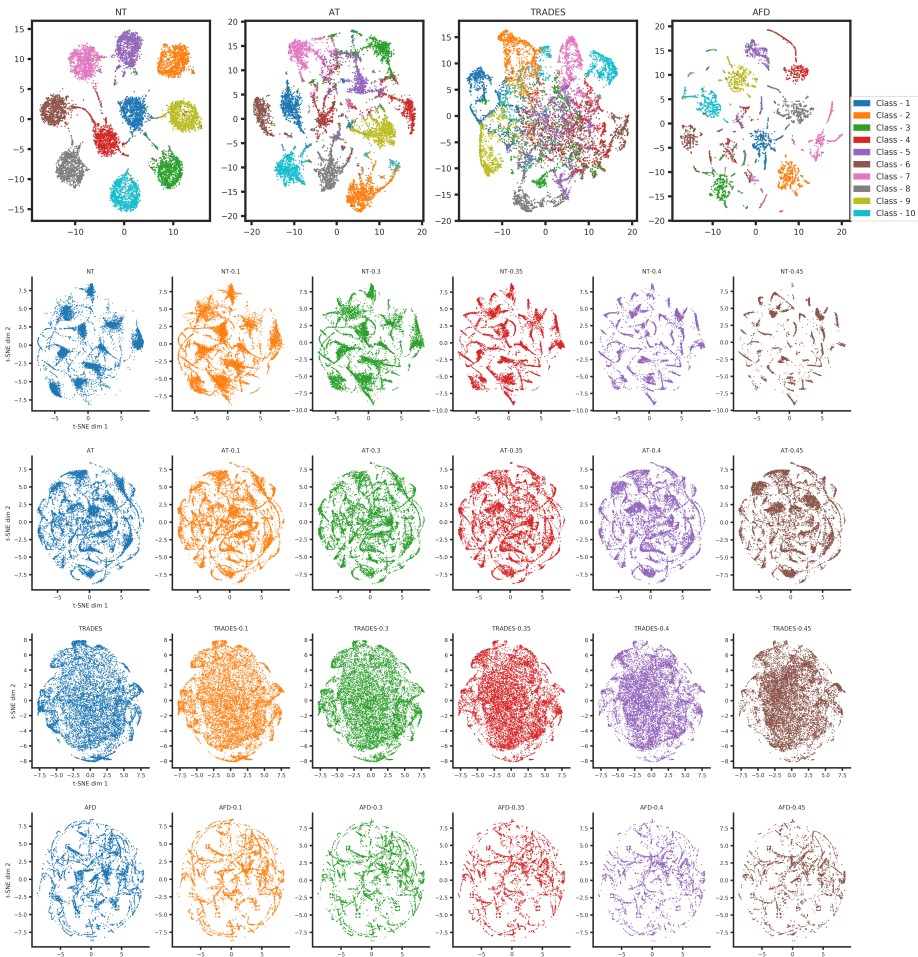

Figure A7: Scatter plot of 2-dimensional t-SNE projection [33] of the representations derived from training the ResNet5 architecture on CIFAR10 dataset. (top row) t-SNE projection of representations of natural images for networks trained with different methods. Each point corresponds to the representations of one of the images from the CIFAR10 test-set. (rows 2 to 5) t-SNE projection of the representations of natural and adversarial CIFAR10 test-set images. Columns are sorted from left to right with the strength of the perturbation (left-most column corresponds to natural images and right-most column with highest tested perturbation). NT: naturally trained; AT: adversarially trained[34]; TRADES: [57];AFD: adversarial feature desensitization.

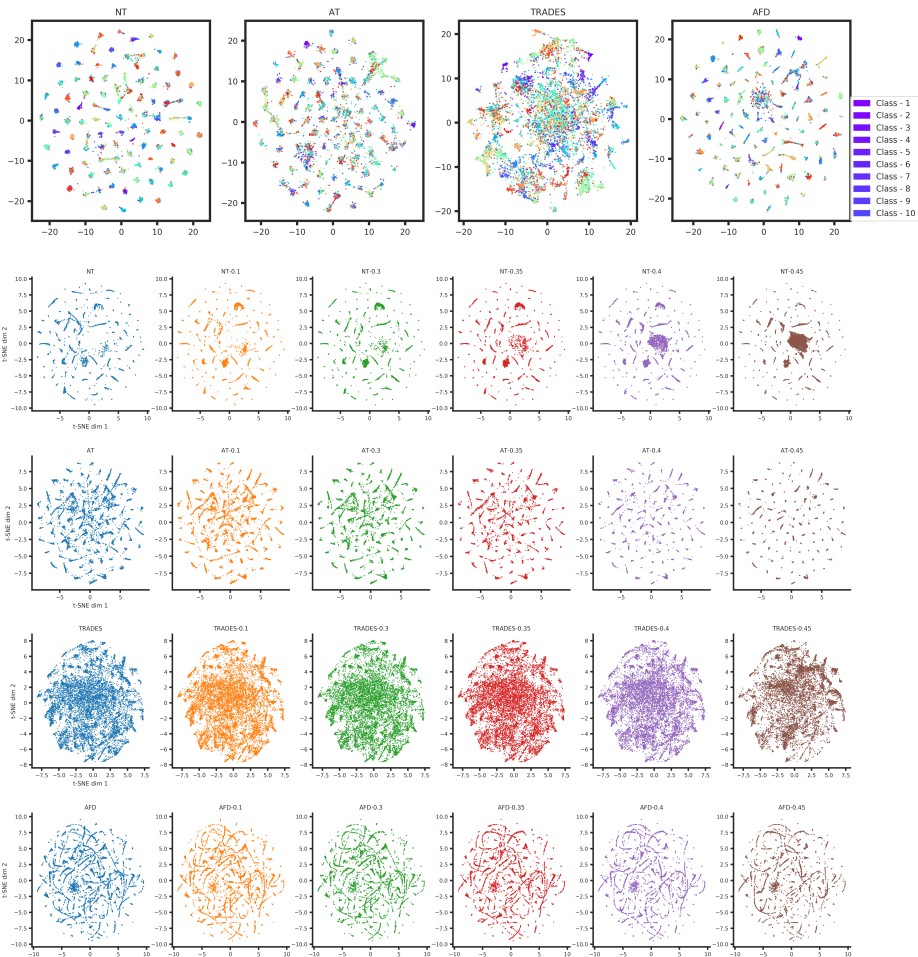

Figure A8: Scatter plot of 2-dimensional t-SNE projection [33] of the representation derived from training the ResNet5 architecture on CIFAR100 dataset. (top row) t-SNE projection of representations of natural images for networks trained with different methods. Each point corresponds to the representation of one of the images from the CIFAR100 test-set. (rows 2 to 5) t-SNE projection of the representation of the natural and adversarial CIFAR100 test-set images. Columns are sorted from left to right with the strength of the perturbation (left-most column corresponds to natural images and right-most column with highest tested perturbation). NT: naturally trained; AT: adversarially trained [34]; TRADES [57]; AFD: adversarial feature desensitization.