# OpenReview forum: "Adversarial Feature Desensitization"
_NeurIPS.cc/2021/Conference — NeurIPS 2021 Poster_

### Official Review · Reviewer_WEvD · 2021-07-07

**Rating:** 5
**Confidence:** 4

**Summary:**

This paper introduced using domain adaptation for adversarial defence and following a similar setup as GAN. The proposed Adversarial Feature Desensitization (AFD) shows promising results on a wide range of datasets.

**Limitations And Societal Impact:**

Authors have adequately addressed the limitations and potential negative societal impact

**Main Review:**

---
Strength:
- Domain adaptation between clean and adversarial as two different domains is a technically sound idea.
- Experiments considered a wide range of datasets that are commonly used to evaluated adversarial defence. The results are promising. Robustness against stronger shows strong results aginst larger $\epsilon$ attacks than adversarial training (AT) models.
- AFD learning a sparse representation is consistent with AT which also encouraged to learn such representation.

---
Concerns and Questions:
- Evaluation against [AutoAttack](https://github.com/fra31/auto-attack) [11]. This paper claimed that this method could generalize to unseen attacks. But based on the evaluation, I do not see such a claim as valid. AutoAttack has evaluated several existing models and found several overestimated their results; for example, the feature-scattering (Zhang & Wang, 2019) uses an idea very similar to this submission. Though it is not guaranteed to be robust to future attacks, one should report the performance of existing strongest attacks to make such a claim.
- What is the overhead of training in addition to generating the adversarial example?
- Discussion on Adaptive Attack (Tramer et al., 2020), in white-box setting, if adversaries have access to the discriminator $D_\psi$, could they generate adversarial examples that are not on the domain as a possible adaptive attack to circumvent the proposed method?


Zhang, Haichao, and Jianyu Wang. "Defense against adversarial attacks using feature scattering-based adversarial training." NeuIPS (2019).

Tramer, Florian, et al. "On adaptive attacks to adversarial example defenses." NeurIPS 2020.

**Time Spent Reviewing:**

3

---

> ### Author Response · Authors · 2021-08-10
> **AutoAttack evaluation, analysis of the computational overhead, and discussion on adaptive attacks**
>
> We thank the reviewer for providing valuable and thoughtful comments on our paper. We hope that our responses have fully addressed all of the reviewer’s concerns, and remain committed to clarifying any further questions that may arise during the discussion period.
>
> **Evaluation on AutoAttack**: Following the reviewer’s suggestion, we additionally evaluated AFD models on AutoAttack and compared its performance with other baselines (AT and TRADES). Briefly, we found that AFD performed better than both baselines on MNIST (AUC 0.90 for AFD vs 0.89 and 0.88 for AT and TRADES), and performed similar or better than AT but slightly worse than TRADES on the other datasets (CIFAR10: AUC 0.24 for AFD vs 0.22 and 0.25 for AT and TRADES; CIFAR100: AUC 0.13 for AFD vs 0.13 and 0.15 for AT and TRADES; Tiny-imagenet: AUC 0.15 for AFD vs 0.14 and 0.16 for AT and TRADES). This shows that AFD also performs competitively against the challenging AutoAttack perturbation. We will update Tables-1 and 2 and include these results there. As AutoAttack already includes the SQUARE attack which is a challenging black-box attack, we did not further test the model against other black-box attacks.
>
> **Computational overhead**: Most notably, compared to adversarial training, AFD requires two additional backward passes during training. Despite this and considering that most of the computational cost during training is associated with generating the adversarial examples, the computational overhead is still not much higher than other baselines. To evaluate the computational cost of our method and to compare it against other baselines, we measured the average per-epoch training time on the CIFAR10 dataset using the same hardware (2 NVIDIA V100 GPUs). We found that the AFD training time is 31% longer than adversarial training and only 4% longer than TRADES. In the light of this result, we argue that the computational cost of our proposed method is not considerably higher than some of the most well-established adversarial robustness methods in the field. Also, at test time, since the network consisting of F and C networks constitutes exactly the same network (ResNet18) that is used in other baselines, the computational cost of the model during test is the same as the other baselines. We will add the empirical runtime comparisons in a new supplementary table and will add the following sentence to the main paper: “On CIFAR10 dataset, AFD training time was 31% longer than adversarial training and only 4% longer than TRADES (Table-A4).”.
>
> **Discussion on Adaptive Attacks**: We thank the reviewer for the valuable suggestion. If the domain discriminator D is made available to the attacker, it is indeed possible to make stronger attacks on the network using adaptive methods such as Tramer et al. (2020). This also triggers the interesting idea that a second level defense method could be introduced to make the network robust against such second order attacks. We believe this could be an interesting future direction of research. We will add this suggestion as a potential future direction to the conclusions and limitations section of our paper.

---

### Official Review · Reviewer_h6S6 · 2021-07-15

**Rating:** 4
**Confidence:** 3

**Summary:**

From the idea of domain-adaptation, the authors design a defence method to learn robust DNNs. The effectiveness is verified by numerical experiments.

**Ethics Review Area:**

["I don’t know"]

**Main Review:**

The basic idea of this paper that is to minimize the divergence between the feature (representation) distributions of adversarial and natural domains is good. But although there is modification from the previous versions (submitted to ICLR 2021), I agree with those reviewers in some aspects.

1. The explanation about the advantage over adversarial training is not very convincing in my point of view.  The upper bound on the adversarial error is not tight and I am not sure why minimizing it results in more attacks. Even in the case the claim is correct, it cannot explain why the proposed method is better for e.g., white-box and known attacks.

2. In numerical experiments, the improvement is only significant for unseen attacks. But for Table 2, I have some worry about other methods, e.g., TRADES. In many literature, TRADES still can get good defence performance for unseen attacks. At least, the drop from using PGD with different norms was not so large. Another drawback is there is still significant accuracy drop from NT for natural images.

3. The motivation for domain-adaption is not very clear. In fact, in their previous version, domain-adaption is not their motivation. I hope the authors could tell the readers the real motivation/thought.

**Time Spent Reviewing:**

2 days

---

> ### Author Response · Authors · 2021-08-10
> **Clarifying the methodology and motivation**
>
> We first thank the reviewer for providing valuable feedback on our paper. Below, we provide detailed responses to each of the comments. Given that no revisions can be made during the review process, we also note in our responses how we are going to update the paper with the new results and discussion points to reflect on the comments and responses. Next, we want to emphasize that the current version of our draft has substantially been improved in terms of the methodological descriptions, results, and discussions compared to its previous versions. We believe that our submission should be evaluated with respect to its current content and regardless of the previous drafts submitted to other venues or comments on those versions that are by and large irrelevant to the current version. We hope that the reviewer finds our responses convincing and considers increasing her/his score to support accepting our paper. We look forward to clarifying any further points that may arise during the discussion period.
>
> **The explanation about the advantage over adversarial training**: In contrast to AT that directly minimizes the adversarial error ($\epsilon_{Z}'$), AFD minimizes an upper bound on the adversarial error as in equation 5. This means that the parameter update on the feature extractor function parameters $\theta$ will likely improve robustness against other adversarial examples with an expected adversarial error that is larger than $\epsilon_{Z}'$ but smaller than the estimated bound from equation 6. We argue that even in the case that the upper bound is not tight, which is the case here, minimizing it would still be a better proxy to minimizing the variable of interest compared to minimizing the lower bound.
>
> **Reported accuracies for TRADES**: We have used the official TRADES code from the author’s repository along with the suggested hyperparameters in training on all datasets. The accuracies under different attacks were also computed using standard public repositories (Foolbox and Advertorch) and were used consistently across all models. Because of these reasons, there was no bias against any of the baselines in our analyses. We want to point out that unlike most other studies, we have evaluated all of the models over a wide range of epsilons and attack types. The difference between our method AFD and other baselines becomes more evident with increasing the attack strength and that might be the reason why the differences appear more pronounced in our reported results. We would appreciate it if the reviewer could point us to the publications where TRADES performance on these attacks are reported and is different from our reported values.
>
> **Relative drop in adversarial accuracy compared to the NT model**: To the best of our knowledge almost all previous defense methods show some level of drop in their performance on natural images and in a similar fashion AFD also shows such drop. Despite that, the amount of drop observed in AFD is consistently similar or better than that in AT and only worse than TRADES on CIFAR100 dataset. For these reasons, we believe that the lower accuracy on the natural images should not be used as a basis of criticizing our proposed method.
>
> **Motivation**: We want to emphasize again that the current draft should be the basis of evaluations and not any previous version of our paper. As it is stated in the introduction and methods section of our paper, our motivation was to learn high-level features that are robust to adversarial perturbations. We reasoned that, as in out of distribution generalisation tasks, we want our model to be robust to a distributional shift between the source domain used during training and the target domain used at test time.

---

> > ### Comment · Reviewer_h6S6 · 2021-08-16
> > **thanks for the explanation**
> >
> > Thanks for the reply. Generally, I like the attempt to find new defense method that differs from adversarial training, which is also raised as the positive part by reviewer SkVe.
> >
> > The main drawback of adversarial training is the trade-off essence. The most obvious part is the significant drop for natural images. So in my opinion, as a new method essentially different from adversarial training should have significant better performance on natural images, while keep similar robustness for adversarial examples. However, this is not the case in the current version (I did not mean adversarial training has good performance on training. Actually the performance is bad and should be improved)
> >
> > When mentioning the last version, I did not to judge based on that. But only want to know the real motivation, i.e., surely there is significant improvement, but still the two are the same work. This is why I want to ask what is the real motivation. I do not think the general statement " learn high-level features that are robust to adversarial perturbations" is convincing.
> >
> > Generally, I would like to keep my score unchanged.

---

> > > ### Author Response · Authors · 2021-08-19
> > > **AFD models have higher accuracy on natural images compared to adversarial training and TRADES**
> > >
> > > Thank you for your comment.
> > >
> > > We agree with the reviewer on the importance of maintaining a good accuracy over natural images while improving the robust accuracy against adversarial perturbations and we would like to highlight that compared to adversarial training (AT), our proposed method performs higher on CIFAR100 (AFD: 58.87% vs. AT: 56.49%) and Tiny-Imagenet (AFD: 47.70 vs. AT: 43.80) datasets while having similar accuracy on MNIST (AFD: 99.14% vs. AT: 99.35%) and CIFAR10 (AFD: 85.95% vs. AT: 85.92%). Similar comparisons are true with respect to the TRADES method where AFD shows higher natural accuracy on CIFAR10 (AFD: 85.95% vs. TRADES: 81.94%) and Tiny-Imagenet (AFD: 47.70% vs. TRADES: 37.70%), similar accuracy on MNIST (AFD: 99.14% vs. TRADES: 99.14%) and slightly lower accuracy on CIFAR100 (AFD: 58.87% vs. TRADES: 60.32%). Because of this, **we believe our results should be viewed as an improvement over prior defense methods and not be counted as a drawback of our method**.

---

### Official Review · Reviewer_xEbL · 2021-07-16

**Rating:** 3
**Confidence:** 5

**Summary:**

The paper proposes Adversarial Feature Desensitization (AFD) to train classifiers robust to adversarial attacks. In particular, AFD trains jointly a feature extractor, a classifier for the original task and a discriminator which distinguishes between natural and adversarial inputs, in order to find features which are effective for classification but not sensitive to adversarial attacks. In the experimental evaluation on four datasets, AFD outperforms standard methods in most of the cases.

**Ethical Concerns:**

No.

**Limitations And Societal Impact:**

The broader impact is not discussed.

**Main Review:**

Strengths
- Using an approach based on domain adaptation to get robust features seems novel.

- Four datasets are used for the experimental evaluations.

Weaknesses
- The main concern is about the experimental evaluations, since the attacks used seem not sufficient and there are hints suggesting the presence of some form of gradient obfuscation. In Figure 2, many attacks do not reduce the robustness of AFD significantly even for very large thresholds, e.g. against PGD with $\epsilon_\infty=0.125 \approx 32/255$ AFD still has around 50% of robustness, which is around what one would expect for ResNet-18 at $\epsilon_\infty = 8/255$. Additionally, in the same setup on MNIST (Figure A1), AFD has high robustness at $\epsilon_\infty=0.5$, for which any starting image can be turned into uniform grey. This is a hint of gradient obfuscation [A].
Stronger methods like AutoAttack [B] (which supports different norms) and MultiTargeted Attack [C] could be used for evaluation.

Overall, the experimental evaluation is not convincing.

[A] https://arxiv.org/abs/1902.06705
[B] https://arxiv.org/abs/2003.01690
[C] https://arxiv.org/abs/1910.09338

**Time Spent Reviewing:**

3

---

> ### Author Response · Authors · 2021-08-10
> **Empirical results argue against gradient obfuscation**
>
> We thank the reviewer for providing valuable and thoughtful comments on our paper. We hope that our responses have fully addressed all of the reviewer’s concerns, and remain committed to clarifying any further questions that may arise during the discussion period.
>
> **Gradient obfuscation**: We understand the reviewer’s concerns about the possibility of gradient obfuscation. However, we believe that our results are not due to obfuscated gradients because of the following reasons [1]. i) For many attacks on all datasets, the model performance continues to decrease with increased epsilon (Figure 2, A1, A2, A3); ii) The iterative perturbations were consistently more successful than single-step ones; iii) Black-box attacks were significantly less successful than white-box attacks (Table-1, A3);
>
> **Evaluation on AutoAttack**: Following the reviewer’s suggestion, we additionally evaluated AFD models on AutoAttack and compared its performance with other baselines (AT and TRADES). Briefly, we found that AFD performed better than all baselines on MNIST (AUC 0.90 for AFD vs 0.89 and 0.88 for AT and TRADES), and performed similar or better than AT but slightly worse than TRADES on the other datasets (CIFAR10: AUC 0.24 for AFD vs 0.22 and 0.25 for AT and TRADES; CIFAR100: AUC 0.13 for AFD vs 0.13 and 0.15 for AT and TRADES; Tiny-imagenet: AUC 0.15 for AFD vs 0.14 and 0.16 for AT and TRADES). This shows that AFD also performs competitively against the challenging AutoAttack perturbation and provides further support against the possibility of gradient obfuscation. We will update Tables-1 and 2 and include these results there.
>
> **The broader impact is not discussed**: We had put the broader impact section in the supplementary materials due to space limitation and following the new guidelines that allows this section to be put outside the main paper.
>
> [1] Athalye et al. "Obfuscated gradients give a false sense of security: Circumventing defenses to adversarial examples." International conference on machine learning. PMLR, 2018.

---

> > ### Comment · Reviewer_xEbL · 2021-08-30
> > **Post rebuttal comments**
> >
> > I thank the authors for the reply and the additional experiments. Below a few comments.
> >
> > - The checks for gradient obfuscation are only guidelines, and in my opinion should be interpreted for each case independently: the robust accuracy indeed decreases in the plots in Figure 2, but the decrements are very small in most of the cases and the curves do not seem to converge to 0% at reasonable values of $\epsilon$. Also for the case of MNIST mentioned above, there's definitely some issue in the evaluation since for $\epsilon_\infty=0.5$ the robustness can't be non-trivial. Table 1 considers transfer-based attacks, while there are much stronger score-based methods as black-box attacks.
> >
> > - It's quite hard to interpret the level of robustness from the AUC since, I assume, only a few thresholds are used. Which are the values at the thresholds used in Table 1?
> >
> > - Thanks for pointing to the "Broader impact" section, I missed it.

---

> > > ### Author Response · Authors · 2021-08-31
> > > **large epsilon robustness and AUC interpretations**
> > >
> > > Thank you for your comments.
> > >
> > > We agree with the reviewer that in the case of large epsilons like $\epsilon=0.5$ and given unlimited resources, the attacker should find an adversarial example for any input pattern. However, please note that in our experimental settings each attack algorithm is given a limited number of iterations and is restricted with respect to the step size it could take on each iteration. Because of this, it is possible for some attack algorithms to not be able to find adversarial examples for many images even with large epsilons. Also, on MNIST dataset when using AutoAttack with $\epsilon\sim0.4$ the accuracy of the AFD model is reduced to 0%. However, AFD accuracy was still higher than baselines for smaller $\epsilon$ values.
> > >
> > > $\epsilon=0.3$: AFD: 90.12% AT: 89.99% TRADES: 88.66%
> > >
> > > $\epsilon=0.35$: AFD: 26.01% AT: 0.05% TRADES: 0.00%
> > >
> > > We used the AUC measure to summarize the robust accuracies for each method over the range of $\epsilon$ values that we had tested. The tested $\epsilon$ values are listed in Table-A2 in the supplementary materials. The figures in the main paper and the supplementary further also show the accuracy values attained for each $\epsilon$ value.

---

### Official Review · Reviewer_SkVe · 2021-07-17

**Rating:** 6
**Confidence:** 5

**Summary:**

This work propose a new formulation for training adversarially robust deep networks, which utilizes the knowledge from the domain-adaptation field. The proposed method of AFD is evaluated on benchmark datasets.

**Main Review:**

1) This paper is well-written. The writing is clear and straightforward. The experimental setup is aligned with other works in this field.
2) This method seems novel to me. I have not read works that proposes a similar understandings or methods.

However, I have four major questions:

1) After reading the paper, I understand the training procedure of AFD, but I am not quite certain about the testing procedure. What exactly will happen during testing? A : Will the GAN part be eliminated during testing while the discriminator of F proceeding predictions? B : Or will the GAN part also participate in the predictions? If the GAN is kept (situation A), AFD clearly bring extra computation expanse, and the author should present the amount of the increased computation complexity. By the way, in algorithm, how exactly does AFD generate xi' from xi? with PGD?

2) Why only present your results on the relatively small model of ResNet18. Also, in Table 1, considering both TRADES and AT present their results on the stronger WRN-34-10/WRN-28-10 but the authors of this paper did not present these results, I suspect the effectiveness of this method may come from the increased GAN part (if the answer of Q1 is situation A). BTW, the attacks evaluated in this work is somewhat out of date and weak. I hope to see evaluations on more powerful attacks like AutoAttack.

3) The major benefit of AFD, as claimed by the authors for several times. is its capability to generalize to unseen attacks during testing. And they try to prove this theory with Table 2. If the theory is correct, a reasonable results should show that AFD exhibit similar performance against the attacks during training (seen attacks) with other methods like AT, and perform much better on unseen attacks. However, from Table 2, it seems that AFD perform better on all kinds of attacks, including the PGD attack used by training. This fact again make me suspect that the presented superiority of AFD may come from the extra computation of GAN. At least, I am not very convinced about the claim that "AFD  can better generalize to unseen attacks"

4) In the paper, the author says:
> Line 182 : In contrast to this setup, our proposed method minimizes a conservative upper bound on the adversarial error and therefore is more likely to generalize to a larger set of unseen attacks, and to stronger versions of previously seen attacks (i.e. ones that generate higher-loss samples in the inner optimization loop).

I do not understand this argument. Please explain more clearly.

I may lower my score if the author can not address all my four concerns.

**Time Spent Reviewing:**

6

---

> ### Author Response · Authors · 2021-08-10
> **AutoAttack evaluations and more clarifications**
>
> We thank the reviewer for providing valuable and thoughtful comments on our paper. We hope that our responses have fully addressed all of the reviewer’s concerns, and remain committed to clarifying any further questions that may arise during the discussion period.
>
> **Test-time procedure**: To state more clearly, AFD consists of a feature extractor F, a classifier C, and a domain discriminator D that are linked according to Fig.1B. After training, only the feature extractor F and classifier C are kept during testing. The network consisting of F and C constitutes exactly the same network (ResNet18) that is used in other baselines and because of that, the computational cost of the model during test is the same as the other baselines.
>
> **How are the perturbations generated?** As we had stated on page 6 in section “Training parameters and baselines”, we used an adaptive version of the PGD attack during training.
>
> **Evaluation on other Architectures:** Our choice of the network architecture was strictly driven by the training time and the computational costs associated with finding the optimal hyperparameters. As it was mentioned in the paper, to find a good set of hyperparameters that would lead to stable training and achieving good performance we performed a grid-search on a range of choices for the hyperparameters using the ResNet18 architecture and a held-out validation set. We found out that the same choices of hyperparameters sometimes lead to instability between the GAN components in AFD when using other architectures and we would have needed to look for a new set of hyperparameters - a task that was not feasible for us at the moment. Instability during training of neural networks with GAN losses is a well-known problem and is commonly remedied empirically by searching for the set of hyperparameters that lead to a stable training. For this reason, we chose to fix the architecture throughout the analyses. In spite of this, we believe that our reported results demonstrate a fair comparison between our method and the baselines as the same architecture was used throughout our experiments.
>
> **AutoAttack Evaluations:** Following the reviewer’s suggestion, we additionally evaluated AFD models on AutoAttack and compared its performance with other baselines (AT and TRADES). Briefly, we found that AFD performed better than both baselines on MNIST (AUC 0.90 for AFD vs 0.89 and 0.88 for AT and TRADES), and performed similar or better than AT but slightly worse than TRADES on the other datasets (CIFAR10: AUC 0.24 for AFD vs 0.22 and 0.25 for AT and TRADES; CIFAR100: AUC 0.13 for AFD vs 0.13 and 0.15 for AT and TRADES; Tiny-imagenet: AUC 0.15 for AFD vs 0.14 and 0.16 for AT and TRADES). This shows that AFD also performs competitively against the challenging AutoAttack perturbation. We will update Tables-1 and 2 and include these results there.
>
> **AFD generalization:** We want to first clarify again that the domain discriminator (i.e. GAN) in our method is only part of the computation during training and not during test. During testing the model consisting of the feature extractor and the classifier which corresponds directly to the ResNet18 architecture (same as in the baselines) is used to perform the classification task. Because of this reason, there is no extra computational cost associated with our method compared to other baselines during testing. While AFD performs better than the baselines on the seen PGD-$L_{\infty}$ attack, the relative improvements are much larger when being compared on other attacks and larger epsilon values (see Figures 2, A1, A2, A3). These empirical results support our statement that AFD indeed generalizes well to unseen attacks.
>
> **Line 182 clarification:** In contrast to AT that directly minimizes the adversarial error ($\epsilon_{Z}'$), AFD minimizes an upper bound on the adversarial error as in equation 5. This means that the parameter update on the feature extractor function parameters $\theta$ will likely improve robustness against other adversarial examples with an expected adversarial error that is larger than $\epsilon_{Z}'$ but smaller than the estimated bound from equation 6.

---

> > ### Comment · Reviewer_SkVe · 2021-08-12
> > **Response to the rebuttal**
> >
> > Thanks for your reply. The rebuttal of the author solve all my questions and concerns.
> >
> > Generally speaking, I think this work is interesting considering it proposes a new defense method that differs from adversarial training. This is valuable considering that the adversarial training is dominating the field of defense, which is somewhat reaching a bottleneck of development. Finding alternatives to adversarial training may help us achieve new breakthrough. It also bring new understandings by discussing adversarial robustness under the domain adaptation field. After explained by the author, I think the improvement may well be solid.
> >
> > However, this work also has obvious limitations. Especially, as confirmed by the author, due to the difficulty of training, this method can not be applied to large models. This certainly limits the feasibility of the method for now and makes me worry about the future development of the proposed method.
> >
> > Thus, considering the high standard of NeurIPS, I think this work still has some works that could be done, especially the extension to large models like WRN-34-10. I will keep my score unchanged but will not object to a decision of rejection.

---

> > > ### Author Response · Authors · 2021-08-13
> > > **Training cost is comparable to prior methods like TRADES**
> > >
> > > We are glad that we have been able to fully resolve all of the reviewer’s questions and concerns. We wanted to further clarify that our evaluations of the computational cost shows that AFD is only 4% more costly than TRADES which is a widely accepted adversarial defense method. **Importantly, this additional computational cost is by no means prohibitive of scaling our proposed method to larger datasets and its cost remains comparable to prior defense methods (e.g. AT and TRADES).** However, this computational budget is still beyond than what is accessible to our research group.

---

### Decision · Program_Chairs · 2021-09-28

**Decision:**

Accept (Poster)

**Comment:**

Thank you for your submission to NeurIPS.  The reviewers and I are in agreement that there are some interesting aspects to the proposed approach.  However, I share the concerns echoed by the reviewers on the "too good" results as presented in the text.  Specifically, as highlighted e.g. by Reviewer xEbL, the presented results simply don't add up in most cases, showing accuracy that doesn't match up with what is possible for deep networks.  While the reviewers attempted to address these concerns, ultimately these weren't that convincing, and I am in agreement with the reviews that the work needs much more review before publication at NeurIPS.

**Consistency Experiment:**

NeurIPS has a long history of experimentation. In 2014, NeurIPS ran an experiment in which 10% of submissions were reviewed by two independent committees to quantify the randomness in the review process. This year, we repeated a variant of this experiment to see how the quality of the review process has changed over time.  This paper was part of the experiment and was therefore assigned to two committees (consisting of reviewers, an Area Chair, and a Senior Area Chair) that reached independent decisions.  If both committees made the same recommendation, this recommendation was followed. If a single committee recommended acceptance, the paper was accepted (with the exception of a few cases in which the other committee identified what we considered a fatal flaw, e.g., an error in a key result).

This copy’s committee reached the following decision: **Reject**

The other committee assigned to the paper recommended **Accept (Poster)**.  You can find the other set of reviews, along with any follow up discussion with the authors here:
https://openreview.net/forum?id=IIo3Ew4v5tk